# A nap to recap or how reward regulates hippocampal-prefrontal memory networks during daytime sleep in humans

Kinga Igloi[1,2,3]*, Giulia Gaggioni[1], Virginie Sterpenich[1,2,3], Sophie Schwartz[1,2,3]

[1]Department of Neuroscience, Faculty of Medicine, University of Geneva, Geneva, Switzerland; [2]Swiss Center for Affective Sciences, University of Geneva, Geneva, Switzerland; [3]Geneva Neuroscience Center, University of Geneva, Geneva, Switzerland

**Abstract** Sleep plays a crucial role in the consolidation of newly acquired memories. Yet, how our brain selects the noteworthy information that will be consolidated during sleep remains largely unknown. Here we show that post-learning sleep favors the selectivity of long-term consolidation: when tested three months after initial encoding, the most important (i.e., rewarded, strongly encoded) memories are better retained, and also remembered with higher subjective confidence. Our brain imaging data reveals that the functional interplay between dopaminergic reward regions, the prefrontal cortex and the hippocampus contributes to the integration of rewarded associative memories. We further show that sleep spindles strengthen memory representations based on reward values, suggesting a privileged replay of information yielding positive outcomes. These findings demonstrate that post-learning sleep determines the neural fate of motivationally-relevant memories and promotes a value-based stratification of long-term memory stores.

*For correspondence: kinga.igloi@unige.ch

**Competing interests:** The authors declare that no competing interests exists.

## Introduction

It is well established that sleep contributes to memory consolidation processes (*Diekelmann and Born, 2010*; *Maquet, 2001*; *Stickgold and Walker, 2013*). In animals and humans memory traces are reinforced through neural reactivation and trimmed through synaptic downscaling. Critically, both neural replay and synaptic downscaling entail the tagging of neural elements coding for specific memories and coincide with the occurrence of specific oscillatory elements in sleep such as sleep spindles, and slow wave activity (*Ringli and Huber, 2011*; *Tononi and Cirelli, 2003*; *Schwindel and Mcnaughton, 2011*; *Wilhelm et al., 2011*; *Káli and Dayan, 2004*). For memory consolidation to be adaptive, information that is critical for survival, such as stimuli with an emotional or rewarding value, should be consolidated in priority during sleep (*Perogamvros and Schwartz, 2012*; *Sterpenich et al., 2009*). Consistent with this hypothesis, behavioral studies in humans suggest that reward may modulate sleep-related memory consolidation (*Fischer and Born, 2009*; *Werchan and Gómez, 2013*; *Oudiette et al., 2013*). At the neural level, animal studies provide evidence for spontaneous coordinated replay of neural activity in memory and reward brain structures during slow-wave sleep (*Lansink et al., 2009*; *Pennartz et al., 2004*), suggesting that the activation of reward circuits during sleep influences neural plasticity. Further, within the rat hippocampus, the firing patterns of CA1 and CA3 neurons encode the high or low reward outcome of items in an associative memory task (*Mckenzie et al., 2014*). In humans, at wake, the activation of the mesolimbic dopaminergic system enhances associative memory through interactions with the hippocampus (*Shohamy and Wagner, 2008*), as well as subjective confidence and feeling of goal attainment during successful retrieval (*Shohamy and Wagner, 2008*; *Adcock et al., 2006*; *Wolosin et al., 2012*).

**eLife digest** Fresh memories are strengthened while we sleep. However, we don't remember every detail of our daily life experiences. Instead, it is essential that we retain information that promotes our survival, such as what we call "rewards" (including food, money or sex) and dangers that we should avoid.

Igloi et al. sought to find out how the human brain picks out important memories to be consolidated during sleep, while discarding irrelevant information. Healthy participants learned series of pictures associated with either high or low rewards. After learning, some of the participants had a nap, while others remained awake. Directly after this and three months later, all the participants returned for a memory test. Igloi et al. found that the highly rewarded pictures were better remembered at both time points (at the expense of lowly rewarded ones), but only for participants who had slept after learning.

Further analysis revealed that distinctive bursts of brain activity occurring during sleep, so-called "sleep spindles", favor the reorganization of memories stored in a region of the brain called the hippocampus, often considered to be the organ of memory.

These findings uncover how sleep enhances long-term memory selectivity thus demonstratethat sleep does not just passively increase the retention of all memories. In the future, this work may inspire educational strategies that combine the careful use of rewards followed by an overnight period of sleep.

Yet, it is unclear whether the replay of rewarded memories during sleep primarily involves the transfer of hippocampal memories to cortical sites (*Frankland and Bontempi, 2005*; *Gais et al., 2007*; *Takashima et al., 2009*), or the strengthening of memory-reward associations, implicating hippocampal-striatal interactions (*Lansink et al., 2009*). Based on these findings and recent theoretical proposals suggesting that sleep may influence associative memory by facilitating the integration of multi-item sequences (*Stickgold and Walker, 2013*; *Ellenbogen et al., 2007*), a critical and unanswered question that we ask here is whether reward values linked to recent memory items control the remodeling of associative knowledge during sleep. Specifically, we test whether reward influences sleep-dependent memory consolidation by promoting associative processes in the hippocampus, ultimately prioritizing long-term retention and subjective confidence for highly (over lowly) rewarded stimuli.

## Results

We designed an associative memory task in which participants learned series of pictures yielding high or low reward outcomes. To assess the influence of sleep and reward, and their potential interaction, memory for these series was tested following a nap or a rest period, and then again 3 months later during the retest session (*Figure 1A*). The task was composed of eight series of six successive pictures each; four series were associated with high reward outcome (HR: dollar coin) and four with low reward outcome (LR: cent coin) all participants received the same total payment. The participants first saw the eight series once during the encoding session (*Figure 1B* and *Figure 1—figure supplement 1A*), followed by the 2-alternative forced choice learning session with feedback during which participants successively selected the next picture in the series among two presented pictures *Figure 1—figure supplement 1B*). Then, the participants either took a 90-min nap (Sleep group, n = 16) or spent an equivalent period quietly awake (Wake group; n = 15), both monitored by polysomnography EEG. After this delay and again three months later, the participants performed an associative memory task on pairs of pictures with different relational distances (direct, inference of order 1 and order 2). They were presented with one image and were then asked which one among two images was part of the same series as the first image (while the other image belonged to a different series) and gave confidence ratings for each answer (*Figure 1C* and *Figure 1—figure supplement 1C*). Functional MRI (fMRI) data were acquired during all experimental sessions and analyzed using SPM8 (see Materials and methods).

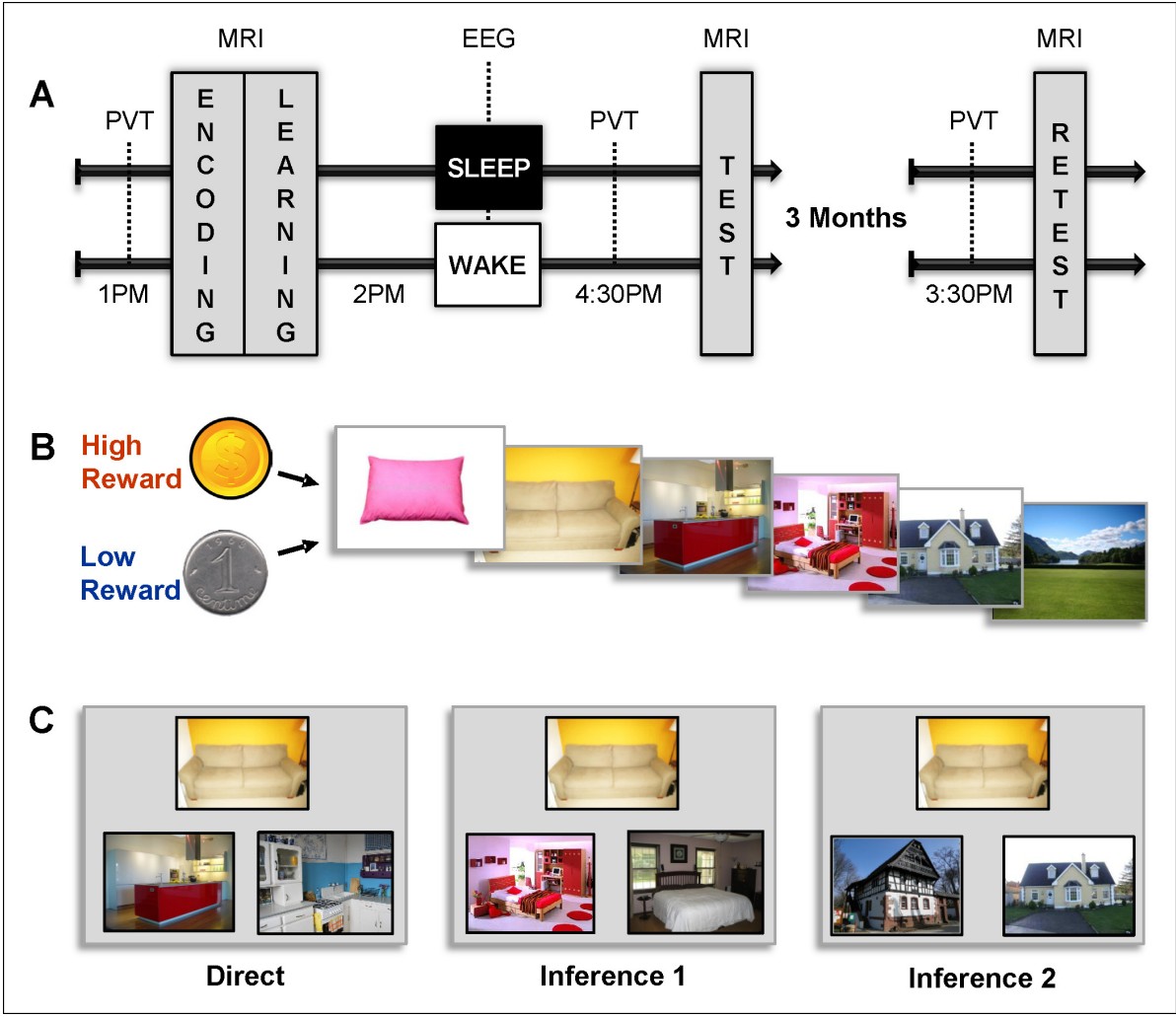

**Figure 1.** Experimental design. (**A**) Overview of the experimental protocol for the Sleep group (upper line) and the Wake group (lower line), composed of three main MRI sessions, each preceded by PVT.Learning and test sessions were performed during one afternoon and separated by a nap (Sleep group) or rest (Wake group) interval monitored by EEG. The retest session occurred three months later. (**B**) Each series of pictures started by a high (dollar symbol) or low (cent symbol) reward cue and was composed of the following series of pictures: pillow, sofa, kitchen, bedroom, house, and landscape. (**C**) Examples of direct trials (left), inference of order 1 trials (middle) and inference of order 2 trials (right). Direct trials were used during the learning, test, and retest sessions, while inference 1 and 2 trials were only used during the test and retest sessions.

The following figure supplements are available for Figure 1:

**Figure supplement 1.** Experimental procedure.

First, we assessed the effect of reward on learning performance (hit rate) during the learning phase using an ANOVA with Reward (high, low) and learning Block (1,2,3, see Materials and methods) as within-subject factors and Group (Sleep, Wake) as between-subject factor. Performance was superior for high reward (HR) than for low reward (LR) picture pairs ($F_{(1,28)} = 35.86$, $p<0.001$), with a significant learning effect over Blocks ($F_{(2,56)} = 20.68$, $p<0.001$), and a Reward x Block interaction effect ($F_{(2,56)} = 4.72$, $p=0.012$). Importantly, there was no Group difference ($F_{(1,28)} = 2.59$, $p>0.05$) and participants of the two groups reached similar levels of performance for HR and LR trials at the third block of the learning (post-hoc test $p>0.05$; *Figure 2—figure supplement 1A*). For all three learning blocks, performance was above chance level (one-way T-tests comparing performance to 50%, all $p<0.05$). Reward-related performance improvement during the learning phase (HR vs. LR) was paralleled by an increase in midbrain activity, in a region compatible with the ventral tegmental area (VTA) [z-score = 3.71 (-3x, -13y, -20z), $p<0.05$, small-volume corrected (SVC) for familywise error

(*Bunzeck and Düzel, 2006*); see Materials and methods] (*Figure 2—figure supplement 1B* and *Supplementary file 2*).

Next, we tested whether the consolidation of rewarded associative memory was selectively enhanced by post-encoding sleep. After a nap (Sleep group) or rest (Wake group), participants were tested on direct pairs of pictures in the series, and on non-consecutive pairs of pictures, that is inference of order 1 and 2 measuring associative memory, namely the strength of the integration of non-consecutive images in a sequence (*Ellenbogen et al., 2007*) (*Figure 1C* and *Figure 1—figure supplement 1C*). Both groups performed above chance level for HR and LR trials and direct, inference 1 and inference 2 trials (one-way T-tests p<0.05). Participants performed better for HR than LR trials (F (1,84) = 15.88, p<0.001), and for close relational distance trials (F(2,84) = 4.16, p=0.018). Further, the Sleep group performed better than the Wake group (F(1,84) = 4.52, p=0.036; *Figure 2A*). Notably, within HR trials, the sleep Group performed better than the wake group (F(1,84) = 5.07, p=0.027), which was not the case for LR trials (F(1,84) = 1.52, p=0.22). However the Group x Reward interaction was not significant (F(1,84) = 0.23, p=0.63). In the Sleep group, the number of slow spindles (11–13 Hz) correlated with memory improvement from learning to test specifically for HR (R = 0.69, p<0.001) and not for LR trials (R = -0.27, p>0.05; *Figure 2B*). The correlation for HR trials was significantly higher than that for LR trials (Fisher's z-score = 2.15, two-tailed p=0.01) (*Lee and Preacher, 2013*). Additionally, the correlation effect was not linked to sleep duration as there was no correlation between performance improvement in the HR trials and sleep duration, while the correlation with performance improvement in HR persisted when considering slow spindles density (number of slow spindles per second of sleep; R = 0.586, p=0.028).

To investigate fMRI responses from the Sleep and Wake groups during the test session, we used a general linear model distinguishing successful trials according to Reward (HR, LR), Relational Distance (direct, inference 1, inference 2) and Confidence ('certain' vs lower confidence judgments, i.e. 'not sure', 'guess', 'by elimination'). A direct comparison between the Sleep and Wake groups for HR versus LR (independently of Relational Distance) revealed increased activity in the right hippocampus [z-score = 4.04 (30x, -10y, -23z), p<0.05 SVC; *Figure 2C*)]. Within this interaction, post-hoc analysis of the extracted beta values showed that both for Sleep and Wake groups HR condition was different from LR condition (both p=0.02). Because the integration of distant associative memories possibly relies on sleep spindles (*Tamminen et al., 2010*), we tested for the brain correlates of this functional relationship. To test specifically for the integration of associative memories (and not the consolidation of previously seen associations, as in direct trials), we considered associations between non-consecutive images with distinct relational distances by comparing inference 2 to inference 1 trials (inference 2 > inference 1). We found that the number of slow spindles in the Sleep group positively correlated with activation in the right hippocampus for HR distant associations [HR inference 2 versus HR inference 1; z-score = 3.75 (27x, -16y, -11z), p<0.05 SVC; *Figure 2D*], and not for LR associations.

We assessed long-term memory in a retest session identical to the test session, three months later. Both Sleep and Wake groups performed above chance level for HR and LR series (one-way T-tests, all p<0.05; *Figure 3A*). While we found no main effect of Group, Reward, or Relational Distance (ANOVA all p>0.05), we found a Group x Reward interaction (F(1,66) = 4.21, p=0.044). Only the Sleep group remembered HR compared to LR series better (T(32) = 2.91, p=0.006), while no such difference was found in the Wake group (T(38) = -0.25, p=0.80). In fMRI, we tested for Group differences between HR and LR trials according to Relational Distance. We observed a selective increase of left hippocampus activity for HR versus LR during inference 2 trials in the Sleep group compared to the Wake group [z-score = 3.78 (-36x, -28y, -8z), p<0.05 SVC; *Figure 3B*], we observed no Group difference for direct and inference 1 trials. Post-hoc analyses of the hippocampal beta values showed that the interaction was due to an increased hippocampal response to HR vs. LR trials in the Sleep group only (p=0.03). Further, using psychophysiological interaction (PPI), we asked whether sleep had some long-term effects on the functional coupling between the hippocampus and other brain regions during the processing of HR (vs. LR) trials (see Materials and methods). Both the caudate nucleus [z-score = 3.41 (15x, 20y, 7z); p<0.05 SVC] and the medial prefrontal cortex [z-score = 2.99 (15x, 59y, -5z); p<0.05 SVC] showed such pattern of increased functional connectivity in the Sleep (compared to the Wake) group (*Figure 3C*). Because striatal activation may be relevant for the reprocessing of rewarded associations during sleep (*Lansink et al., 2009*), we tested whether the strength of the caudate-hippocampal functional coupling correlated with the number of slow

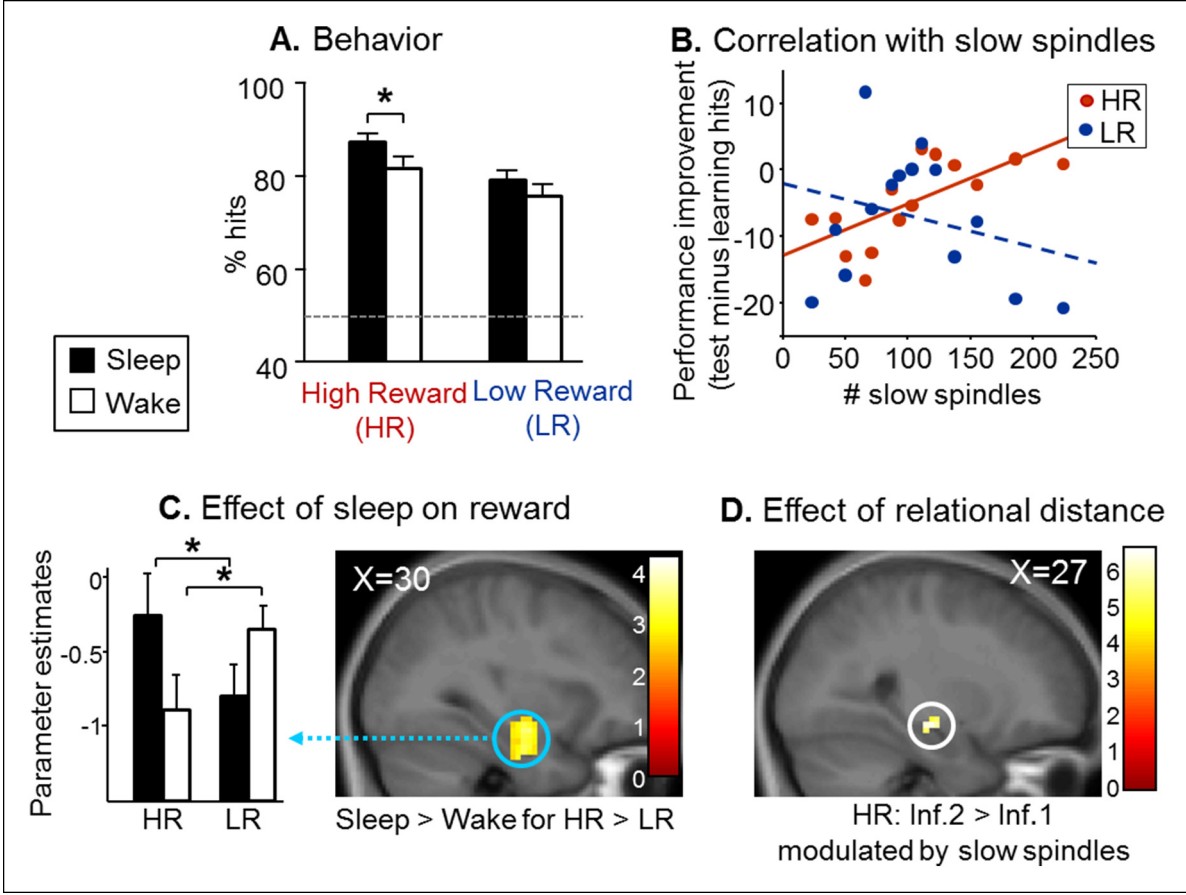

**Figure 2.** Test results. (**A**) Better performance for the Sleep group than the Wake group and also for High than Low reward trials. (**B**) Memory improvement for HR series correlated with the number of slow spindles. (**C**) Increased right hippocampal activity for HR than LR for the Sleep compared to the Wake group. (**D**) Increased right hippocampal response for HR inference of order 2 compared to inferences of order 1 correlated with the number of slow spindles for the Sleep group. All activation maps are displayed on the mean T1 anatomical scan of the whole population. For display purposes, hippocampal activations are thresholded at p<0.005.

The following figure supplements are available for Figure 2:

**Figure supplement 1.** Learning results.

**Figure supplement 2.** Behavioral results at test.

sleep spindles during the nap in the Sleep group, and found that this correlation was significant (R = 0.591, p=0.028, *Figure 3D*). Note that no such correlation was observed for the connectivity between the hippocampus and medial prefrontal cortex (mPFC).

Beyond performance effects, it has been shown that remembering emotional events elicits a stronger feeling of recollection (*Sharot et al., 2004*), yet higher confidence seems adaptive only if associated with accurate memory recall (*Schultz, 2006*). We therefore analyzed 'certain' confidence responses. To correct for potential performance biases, we considered the percentage of 'certain' responses for hits only. For the test session, we found a main effect of Group (F(1,78) = 4.90, p=0.029; more 'certain' responses for the Sleep group), of Reward (F(1,78) = 59.06,p<0.001; higher confidence for HR (*Figure 4A*). Further, we found a Group effect (Sleep>Wake) within HR hits (F (1,78) = 6.63, p=0.01), and not for LR hits (F(1,78) = 0.73, p=0.39). Importantly, the Group effect was not due to an unspecific influence of sleep, as confidence judgments for incorrect responses did not differ between the groups, neither for HR trials (U = 72.50, p=0.35) nor for LR trials (U = 84, p=0.51). Note that we performed U-Tests because of small sample size and non-normal distribution of incorrect responses. For the retest session, we found a main effect of Reward (higher confidence

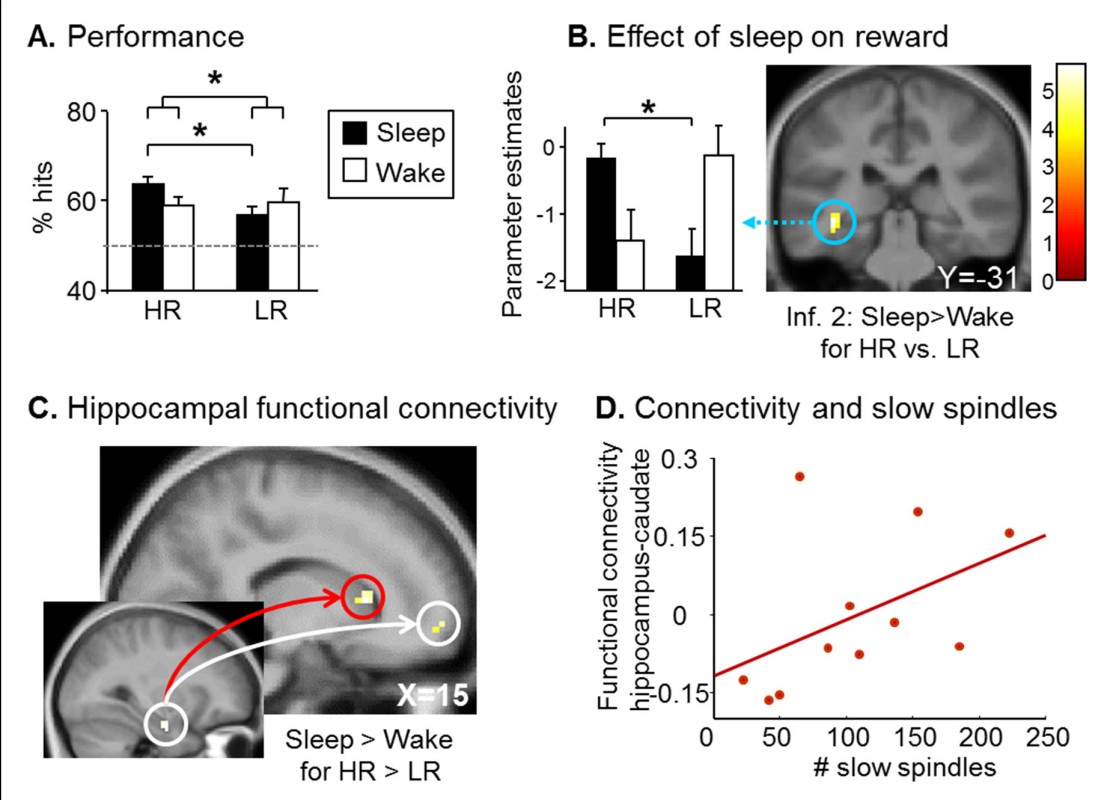

**Figure 3.** Retest results. (A) The Sleep group performed better for HR trials than for LR trials. (B) Increased left hippocampus activity during the retest session for HR vs. LR for the Sleep compared to the Wake group, selectively during inference 2 trials. (C) PPI for the retest session, using the seed in the right hippocampus from *Figure 2C*. Increased functional coupling with the caudate nucleus and the medial prefrontal cortex during HR vs. LR trials, selectively for the Sleep group compared to the Wake group. (D) Beta values of the PPI around the caudate nucleus peak correlated with the number of slow sleep spindles. Activation map displayed on the mean T1 anatomical scan of the population. For display purposes, hippocampal activations are thresholded at p<0.005.

The following figure supplements are available for Figure 3:

**Figure supplement 1.** Detailed retest behavioral results.

for HR trials; F(1,66) = 18.92, p<0.001) and a post-hoc difference between HR and LR confidence within the Sleep group only (p=0.003 for the Sleep group and p=0.38 for Wake group) (*Figure 4B*). At retest, there were very few certain responses. To obtain correctly sized samples to analyze confidence judgments at retest, we grouped 'certain', 'by elimination' and 'not sure' answers together and contrasted them to 'guess' answers. In line with our main imaging results for inference 2 trials (*Figure 3B*), this analysis on confidence yielded higher activation in the parahippocampus [z-score = 3.49 (27x, -25y, 26z) p<0.05 SVC] and in the putamen [z-score = 3.46 (-18x, -1y, 7z), p<0.05 SVC] for Sleep > Wake selectively for HR inference 2 when comparing some confidence to guess (*Figure 4C*).

## Discussion

Here we show that sleep favors the selectivity of memory consolidation and confidence for these memories by promoting the integration and long-term retention of the most important (i.e. here highly rewarding) memories.

First, our results show that slow sleep spindles were critically involved in the consolidation of series with high reward outcome (*Figure 2B*), consistent with Schmidt et al. (*Schmidt et al., 2006*) who reported that memory improvement for word-pairs over a nap implicated slow spindles. Sleep spindles may induce long-term synaptic plastic changes (*Marshall et al., 2006*), thus consolidating newly learned information into a more stable form of long-term memory. Slow spindles (11–13 Hz)

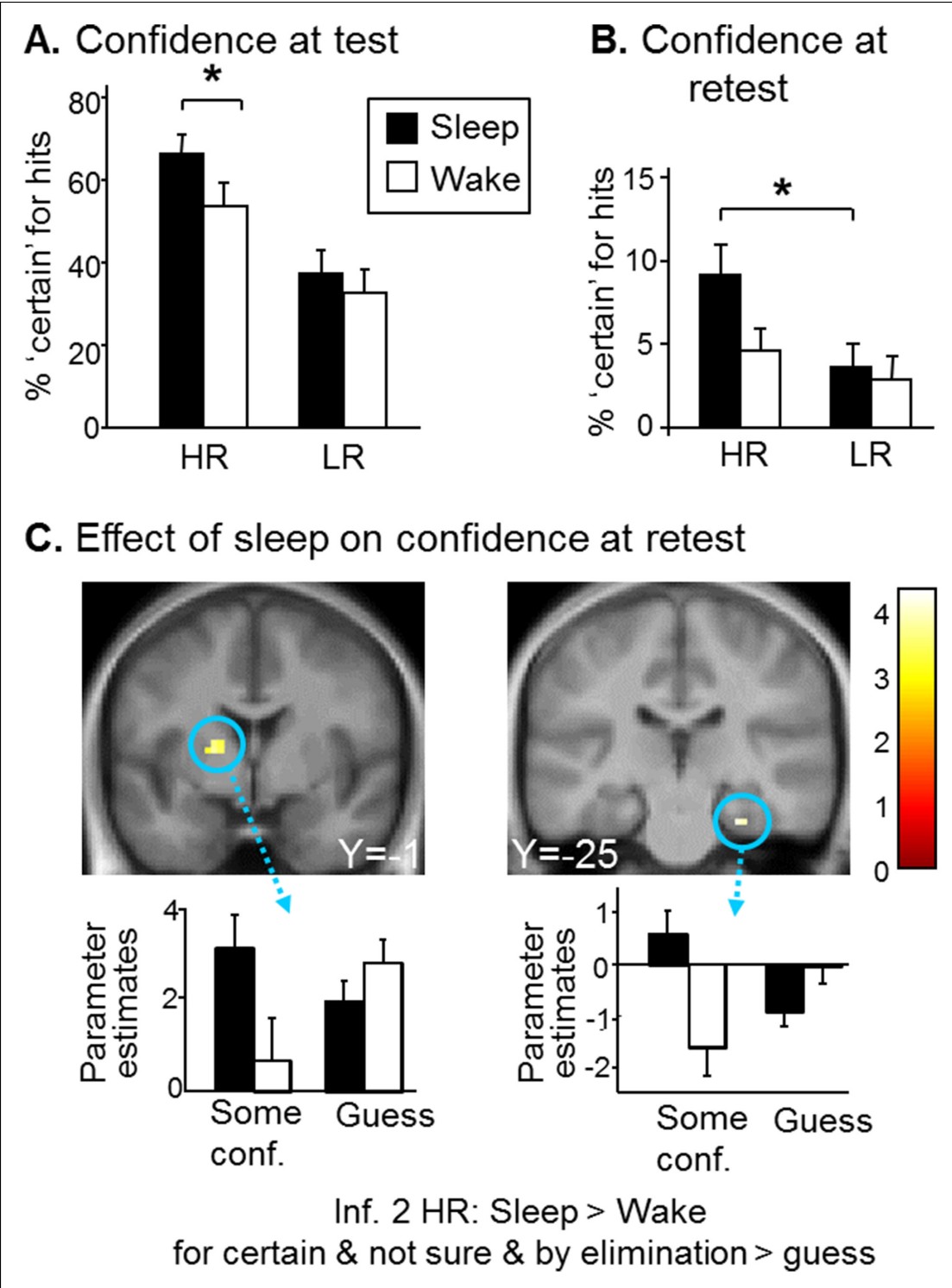

**Figure 4.** Confidence results. (**A**) Higher confidence on hits for the Sleep group than the Wake group and for HR vs. LR trials during the test session. (**B**) Higher confidence on HR vs. LR hits for the Sleep group during the retest session. (**C**) Greater activation of the left caudate and right hippocampus for the 'certain & by elimination & not sure' vs 'guess' confidence judgments for inference 2 HR trials in the Sleep compared to the Wake group during the retest session. Activation map displayed on the mean T1 anatomical scan of the population.

are dominant over the prefrontal lobe (*Saletin and Walker, 2012*) and are related to a cross-linking of transferred information within prefrontal circuitry (*Clemens et al., 2007*). Moreover, activity in dopaminergic reward regions such as the VTA is increased during slow spindles in humans (*Schabus et al., 2007*). Consistent with these observations, we found that slow spindles after learning were related to the functional interplay between the hippocampus and the caudate nucleus during the long-term recall of high rewarded series. From these connectivity findings, we conclude that slow spindles may primarily favor the long-term consolidation of rewarded stimuli across striatal and hippocampal networks. We may speculate that enhanced hippocampal recruitment for HR trials at retest could possibly foster recall processes subserved by the mPFC, a brain region reportedly involved in the retrieval of episodic memories (*Düzel et al., 1999*).

Second, our results confirm that sleep does not only strengthen memory for recently (and possibly more strongly) encoded items (*Diekelmann and Born, 2010*) but also boosts the integration of associative memories, via a hippocampus-dependent mechanism (*Werchan and Gómez, 2013*; *Ellenbogen et al., 2007*; *Lau et al., 2010*) (*Figure 2D*). Importantly, there was no difference between reaction times for inference trials and direct trials (*Table 1*), supporting the idea that the consolidation of associative memory involved the integration of discrete events (i.e., generalization of memories) rather than sequential inferential reasoning (*Shohamy and Wagner, 2008*). This mechanism could promote the conversion of implicit forms of memory into more explicit and conscious memories (*Wilhelm et al., 2013*), and also facilitate the access to remote associations (*Zeithamova et al., 2012*).

Third, we show that sleep after learning reinforces the subjective feeling of recollection for correct recall (*Figure 4A*), and at long-term retest selectively enhances performance for rewarded information (*Sharot et al., 2004*) (*Figure 3A*), although this interaction effect was not present at short-term. These effects were not present for missed trials and were thus not due to a general increase in confidence judgments after sleep. Moreover, we report that medium to high confidence is characterized by parahippocampal and striatal activation at retest. Both these structures have previously been linked to confidence measures (*Daniel and Pollmann, 2012*; *Eichenbaum et al., 2007*), with parahippocampal activation involved in remembering contextual information about a memory, which contributes to increased confidence about the information being recalled (*Eichenbaum et al., 2007*), while striatum (in particular putamen) activation has been shown to reflect an internal error signal about probable outcome in the absence of feedback (or prediction error of confidence measures) (*Daniel and Pollmann, 2012*).

At a more conceptual level, rewards may act as a 'relevance tag' that would prioritize the neural reprocessing of associative memories during sleep via a 2-step process: (i) during encoding, potential rewards activate dopaminergic midbrain regions, which would seal a 'relevance tag' to recently learned and rewarded information; (ii) during sleep (in particular sleep oscillations), the activation of a striatal-hippocampal network would favor the reprocessing of recent memories with a high relevance (*Perogamvros and Schwartz, 2012*).

Overall, by determining the fate of motivationally-relevant memory traces, post-learning sleep, even in the form of a nap, sharpens the skyline of our memories.

**Table 1.** Reaction times (mean ± SEM) for the test phase.

| | | HR | | | LR | | |
|---|---|---|---|---|---|---|---|
| | | Direct | Inference 1 | Inference 2 | Direct | Inference 1 | Inference 2 |
| Sleep | Mean | 939.83 ± 19.16 | 1144.29 ± 21.51 | 1167.24 ± 22.35 | 1207.46 ± 21.36 | 1434.43 ± 22.69 | 1318.69 ± 22.61 |
| | Median | 752.07 ± 18.59 | 1006.20 ± 21.39 | 1024.83 ± 21.86 | 1066.83 ± 20.12 | 1385.43 ± 23.82 | 1132.13 ± 22.21 |
| Wake | Mean | 1164.14 ± 19.18 | 1353 ± 27.81 | 1405.74 ± 21.91 | 1220.55 ± 20.87 | 1305.55 ± 21.74 | 1290.21 ± 26.50 |
| | Median | 994.17 ± 20.83 | 1226.83 ± 29.18 | 1261.27 ± 22.99 | 1067.90 ± 21.86 | 1197.50 ± 23.13 | 1231.40 ± 27.78 |

## Materials and methods

### Participants

Thirty-one healthy young volunteers (16 women and 15 men, age range = 18–30 years old) gave written informed consent and received financial compensation for their participation in this study, which was approved by the Ethics Committee of Geneva University Hospitals. All participants were right-handed, non-smokers, free from psychiatric and neurological history, and had a normal or corrected-to-normal vision. They were within the normal ranges on self-assessed questionnaires for depression, anxiety, circadian typology, had no excessive daytime sleepiness and reported taking regular naps. Before inclusion in the study, all selected participants came for a habituation nap monitored by polysomnography. They then kept a regular sleep-wake schedule during five days prior to the experimental day. Compliance was documented by actigraphy (Actiwatch, Cambridge Neuroscience, Cambridge, UK) and sleep diary. Moreover, they were requested to refrain from all caffeine and alcohol-containing beverages and intense physical activity for the 48h preceding the experiment. Participants were randomly assigned to either a 'Sleep' group (n = 16, 8 men), or to a 'Wake' group (n = 15, 7 men). There was no group difference for any of the questionnaires (all p>0.05), so both groups had in particular equal sensitivity to reward as assessed by the SPSRQ questionnaire.

### Experimental procedure

Participants arrived at 12:45 PM at the Brain and Behavior Laboratory of the University of Geneva. Before each fMRI session, participants got acquainted with the task on two series of pictures that were not used in the main experiment. At 1 PM, the participants were comfortably installed in the fMRI scanner, and performed the encoding session directly followed by the learning session (*Figure 1A*). Then, electrodes were applied to all the participants to ensure similar experimental conditions for all participants: between 2 PM and 3:30 PM participants of the Sleep group took a nap and participants of the Wake group stayed quietly awake in a sound-attenuated room. At 4:30 PM, participants underwent the test fMRI session. A surprise retest session similar to the test session took place three months later at 3:30 PM. Before each fMRI session, a psychomotor vigilance task (PVT) was performed (*Blatter and Cajochen, 2007*) (see *Table 2*).

*Behavioral task.* We developed an associative memory task comprising 8 series of pictures; 4 series yielded high reward outcome (HR) (1 dollar symbol for each picture correctly selected) while 4 series yielded low reward outcome (LR) (1 cent symbol). Each series was composed of 6 photographs presented in the following order: pillow, sofa, kitchen, bedroom, house, and landscape (*Figure 1B*). We told the participants that their final gain depended on their performance, namely that the dollar sign indicated a sequence that would be more rewarded while the cent indicated a sequence proportionally less rewarded. Critically, we also explicitly told them that both the dollar and the cent were 'symbols' and that we would convert their performance (not the accumulated US tokens) into Swiss Francs at the end of the learning session, so that if they performed well they would receive the maximal amount (i.e. 120 Swiss Francs). This manipulation was meant to (i) ensure that the participants paid attention to the distinct reward levels associated with each sequence, (ii) increase the participants' motivation to obtain the rewards, and (iii) also ensure that each participant got the same final monetary outcome (i.e. 120 Swiss Francs). Participants were also informed in the initial instructions that they would have to do an inference task, and were explicitly asked to memorize series as wholes rather than as isolated pair-wise associations.

**Table 2.** Psychomotor vigilance task results (mean ± SEM).

|  | PVT 1 | PVT 2 Sleep group | PVT 2 Wake group | PVT 3 |
|---|---|---|---|---|
| Mean RT | 264.92 ± 4.42 | 263.29 ± 4.45 | 256.44 ± 4.74 | 267.67 ± 4.37 |
| Median RT | 255.02 ± 3.95 | 250.81 ± 3.64 | 249.38 ± 4.30 | 258.12 ± 4.21 |
| False alarms | 0.42 ± 0.95 | 0.55 ± 0.94 | 1.15 ± 1.41 | 0.4 ± 0.89 |
| Lapses | 0.42 ± 0.8 | 0.72 ± 1.02 | 0.38 ± 0.70 | 0.4 ± 0.7 |
| Mean RT fastest 10% | 215 ± 3.59 | 204.93 ± 4.48 | 208.69 ± 4.80 | 213.50 ± 4.63 |

*Encoding session.* Participants watched each of the 8 series once, one picture at a time (2000 ms each), and were asked to encode them. At the beginning of each series, a dollar or a cent picture indicated the reward value (high or low) associated with the series (*Figure 1B* and *Figure 1—figure supplement 1A*).

*Learning session.* Each series is formed by 6 pictures, which that can therefore be grouped into 5 successive direct pairs (pillow-sofa, sofa-kitchen, kitchen-bedroom, bedroom-house, house-scene). Participants were trained on all successive pairs of each of the 8 series in the following way : participants were shown the first picture (pillow) of the series. Then, the same picture was shown together with two options for the second item (sofa). The onset of this display was used for the fMRI analysis (red frame on *Figure 1—figure supplement 1B*). Two seconds later, when the sentence 'choose next picture' appeared in the middle of the screen, participants could give their answer by pressing on an MRI compatible response box (Current Designs Inc., USA) (*Figure 1—figure supplement 1B*). Next, the correct second item (sofa) was presented for two seconds, followed by this item together with the two options for the third item (kitchen), and so on until the sixth picture (landscape). At the end of each series, participants were shown how much they earned in dollars or cents, for high or low reward outcome series respectively. During the learning, participants got feedback about the correctness of their choice on each trial: after each 'choice' display, the correct picture was presented on the subsequent trial, as the next element in the sequence for which the following picture should be selected, and so on until the last trial, for which the ultimate correct choice (landscape) was also shown. The total amount of correct responses on a given sequence was also summarized at the end of the sequence, as a display with filled dollar (for HR) or cents (for LR) symbols (see *Figure 1—figure supplement 1*). All volunteers underwent 3 blocks of learning; in each block, each of the 8 series was presented once, in a randomized order.

*Nap time.* Half of the participants took a nap (Sleep group), the other half stayed awake (Wake group), both for 1h30. The Wake group was allowed to read in dim light (25 lux), on a topic not involving memorization or high cognitive load. For both groups the temperature of the room was controlled (21°C) and polysomnographic data was continuously recorded (see below EEG acquisition). Participants in the Sleep group were allowed to sleep for up to 90 min and were woken only from sleep stages 1, 2 or REM (see *Supplementary file 1* for characteristics of the nap period for participants in the Sleep group). Before starting the test phase, the participants spent at least 40 min awake, in order to dissipate effects of sleep inertia in the Sleep group, and they all completed the St. Mary's Hospital questionnaire. One participant's sleep data were lost due to a computer failure. This participant was excluded from the sleep data analyses.

*Test session.* Participants were tested on all possible direct (or immediately consecutive) pairs of pictures in the series, and also on pairs of non-consecutive pictures, i.e., inference of order 1 and 2 pairs. In inference trials, participants were also presented with a cue picture and had to select which of two pictures belonged to the same series as the cue picture, but for pairs of pictures that were more distant in the series (i.e., separated by one or two intermediate pictures; *Figure 1C* and *Figure 1—figure supplement 1C*). During test and retest sessions all possible combinations were presented once per session: all 5 direct pairs (from each of the 8 series), inference 1 pairs (for each of the 8 series there are 4 possible inference 1 pairs: pillow-kitchen, sofa-bedroom, kitchen-house and bedroom-scene) and inference 2 pairs (for each of the 8 series there are 3 possible inference 2 pairs: pillow-bedroom, sofa-house, kitchen-landscape). The order of the presentation of the pairs was randomized. Like in the learning session, they were first shown one picture alone, then the 2-alternative forced choice display, which was used as onset time for the fMRI analysis (indicated by a red frame on *Figure 1—figure supplement 1C*). They could only give their answer after a 2 s delay when the 'choose next picture' sentence appeared. After each response, participants rated how confident they were when selecting the correct picture among 4 possible options: 'certain', 'not sure', 'guess', or 'by elimination' (*Figure 1—figure supplement 1C*). In order to prevent further learning, no feedback at all was shown during the test and retest sessions. Two participants (one of the Sleep and one of the Wake group) did not understand the confidence question, the data of these participants wer excluded from the confidence analysis.

*Retest session.* Three months after the experiment, the participants were asked to come back for a retest session that was similar to the test session. Participants did not know at test that there would be a retest session. Out of the 31 participants 25 were able to come back after three months for the retest session.

**Table 3.** Number of correct responses during the three learning blocks (mean ± SEM). Only pairs that were selected correctly 2 or 3 times were considered learnt.

|  | HR (20 direct pairs) | LR (20 direct pairs) |
|---|---|---|
| 3 correct | 15.60 ± 0.48 | 12.93 ± 0.55 |
| 2 correct | 3.37 ± 0.39 | 5.00 ± 0.45 |
| 1 correct | 0.83 ± 0.17 | 1.57 ± 0.29 |
| 0 correct | 0.20 ± 0.09 | 0.50 ± 0.14 |

## Behavioral data analysis

During the learning session, a picture in a series was considered learnt when the participant had selected the correct picture at least twice out of three times during the three learning blocks. Pictures that did not meet this learning criterion were removed from the analysis of the test and retest sessions (3.1 ± 2.26 pairs per participant out of 40, see *Table 3*). One participant had to be excluded of all analyses because of memory performance below two standard deviations of the group mean, thus the final group comprised 15 participants for both Sleep and Wake groups, which were included in fMRI analyses.

All behavioral analyses were performed using Statistica (Version 11, www.statsoft.com, StatSoft, Inc. TULSA, OK, USA); non parametric tests were used when normal distribution and equal variance criteria were not met. Post-hoc tests were performed using the Scheffe method.

## Reaction times analysis

We first removed false alarms (reaction times under 50ms) and then computed the mean and median reaction times values. We did not remove slow reaction times as this was not a rapidity task, and participants were told that they could take up to 8 s to answer. We performed ANOVAs on mean and median RT values with Reward (high, low) and Relational Distance (direct, inference 1, inference 2) as within-subject factors, and Group (Sleep, Wake) as between-subject factor. These analyses revealed a main effect of Reward for both means and medians ($F_{(1, 84)} = 4.8263$, $p=0.031$ and $F_{(1, 84)} = 6.8350$, $p=0.011$, respectively) but no main effect of Relational Distance and no interaction with Relational Distance. Participants were faster for trials belonging to HR series as compared to LR series, thus attesting an influence of the reward manipulation on behavior, see *Table 1*.

Similar analyses on the mean and median reaction times for 'certain' responses during the test phase did not show any main effect of Reward, Relational Distance, or Group, see *Table 4*.

## Psychomotor Vigilance Task (PVT) analysis

PVT was administered three times: before the encoding session (PVT 1), before the test session (after the sleep/rest period; PVT 2), and before the retest session (PVT 3; see *Figure 1A*). Analysis of the PVT data showed that reaction times were normally distributed for Sleep and Wake groups during the learning, test and retest sessions. Importantly, there was no group difference for reaction times, false alarms, or lapses at any time point (all p>0.05), see *Table 2*.

**Table 4.** Reaction times (mean ± SEM) for the certain answers of the test phase.

|  |  | HR | | | LR | | |
|---|---|---|---|---|---|---|---|
|  |  | **Direct** | **Inference 1** | **Inference 2** | **Direct** | **Inference 1** | **Inference 2** |
| Sleep | Mean | 727.98 ± 18.52 | 876.96 ± 19.29 | 882.20 ± 20.68 | 609.91 ± 21.69 | 952.90 ± 21.21 | 721.28 ± 23.30 |
|  | Median | 665.43 ± 17.20 | 792.3 ± 18.77 | 770.67 ± 19.71 | 553.5 ± 21.06 | 861.167 ± 21.05 | 682.93 ± 22.71 |
| Wake | Mean | 844.88 ± 18.21 | 1064.95 ± 24.45 | 1095.13 ± 27.81 | 686.73 ± 24.63 | 881.59 ± 21.59 | 767.54 ± 22.88 |
|  | Median | 743.23 ± 18.11 | 1086.60 ± 28.67 | 1068.27 ± 27.28 | 637.33 ± 23.90 | 712.17 ± 20.62 | 651.80 ± 22.05 |

## EEG data acquisition and analysis

Nap-time was monitored using a V-Amp recorder (Brain Products, Gilching, Germany). Standard polysomnography included 6 EEG (Fz, Cz, Pz, Oz, C3, C4, reference on both mastoids), chin EMG, and vertical and horizontal EOG recordings (sampling rate: 250 Hz).

For PSG analyses, we used FASST (fMRI Artifact rejection and Sleep Scoring Toolbox; Cyclotron Research Centre, University of Liège, Belgium) implemented in Matlab (MATLAB version 7.13.0.564 R2011b, Natick, Massachusetts: The MathWorks Inc., 2011). Fourteen naps and fifteen periods of quiet wakefulness were visually scored on a 20 s epoch basis by two independent scorers, according to standard criteria by the AASM Manual for the Scoring of Sleep (*Iber et al., 2007*). Additionally, automatic detection of spindles was performed. Sleep spindles were detected based on an algorithm previously developed by Molle et al. (*Mölle et al., 2002*). Sleep spindles were separated according to their main frequency (i.e. maximum power amplitude within the 11–15 hz range), in line with the global standards used in sleep research (*Schabus et al., 2007*; *Maquet, 2010*). Slow sleep spindles were defined as spindles with predominant frequency between 11 and 13 Hz and fast spindles between 13.1 and 15 Hz. Spindle frequency computation was done by a Matlab toolbox (FASST, Cyclotron Liège) (*Schabus et al., 2007*; *Sterpenich et al., 2007*). The distribution of fast and slow spindles has been reported to be bimodal (*Astill et al., 2014*) and corresponds to an approximate topographical localization, fast spindles having a predominant distribution over parietal regions and slow spindles over prefrontal regions (*Schabus et al., 2007*; *Andrillon et al., 2011*). Slow spindles had a predominant frequency below 13 Hz (mean: 12.55; SEM: 0.08) and fast spindles had a predominant frequency above 13.5 Hz (mean: 14.33; SEM: 0.06). Sleep data are summarized in *Supplementary file 1*. None of the participants in the Wake group fell asleep.

## Functional MRI data acquisition and analysis

MRI data were acquired on a 3 Tesla MRI scanner (SIEMENS Trio System, Siemens, Erlangen, Germany). Multislice T2*-weighted fMRI 2D images were obtained with a gradient echo-planar sequence using axial slice orientation (36 slices; voxel size, $3.2 \times 3.2 \times 3.2$ mm; repetition time (TR) = 2100 ms; echo time (TE) = 30 ms; flip angle (FA) = 80°, FOV = 205 mm).

A whole-brain structural image was acquired at the end of the test part with a T1-weighted 3D sequence (192 contiguous sagittal slices; voxel size, $1.0 \times 1.0 \times 1.0$ mm; TR = 1900 ms; TE = 2.27 ms; FA = 9°). An additional structural image was acquired with a proton-density weighted sequence (20 axial slices; voxel size, $0.8 \times 0.8 \times 3.0$ mm; TR = 6000 ms; TE = 8.4 ms; FA = 149°). This acquisition served for the localization of the VTA (*D'Ardenne et al., 2008*). All stimuli for fMRI were designed and delivered using a MATLAB Toolbox (Cogent 2000, http://www.vislab.ucl.ac.uk/cogent_2000.php).

Functional images were analyzed using SPM8 (Wellcome Department of Imaging Neuroscience, London, UK). This analysis included standard preprocessing procedures: realignment, slice timing to correct for differences in slice acquisition time, normalization (images were normalized to an EPI template), and smoothing (with an isotropic 8-mm FWHM Gaussian kernel). A general linear model (GLM) approach was then used to compare conditions of interest at the individual level and then these contrasts from each participant entered a second-level random-effects analysis. All group comparisons were performed using ANOVAs. Correction for multiple comparisons was performed by submitting all reported activations to small-volume correction (SVC) for familywise error (p<0.05) using regions of interest based on the Anatomy toolbox of SPM8 for the hippocampus (SPM Anatomy toolbox 2.1, Forschungszentrum Jülich GmbH), the automated anatomical labeling (aal) atlas for the caudate nucleus, the parahippocampus and the putamen (*Tzourio-Mazoyer et al., 2002*), and using the coordinates from (*Bunzeck and Düzel, 2006*) for VTA-compatible midbrain regions and from (*Düzel et al., 1999*) for the medial prefrontal cortex. Coordinates of brain regions are reported in MNI space.

## Functional MRI analysis of the learning phase

To investigate fMRI responses from the learning session, we first used a general linear model at the single individual level including 3 sessions for the 3 learning blocks. Each session contained 2 regressors for successful trials and 2 regressors for misses according to their reward outcome (HR, LR) and the 6 motion parameters derived from the spatial realignment added as covariate of no interest. We

then performed a contrast between HR and LR hits for each participant and entered the resulting statistical maps into a second-level 2-sample t-test which also included the difference in performance between HR and LR series (i.e., HR minus LR performance) as a covariate. This analysis allowed us to identify brain regions selectively contributing to reward-related performance improvement. This analysis revealed increased activation in a midbrain region compatible with the VTA [z-score = 3.71 (-3x, -13y, -20z), $p < 0.05$, small-volume corrected (SVC) for familywise error] using the VTA coordinate from Bunzeck and Duzel (*Bunzeck and Düzel, 2006*; *Figure 2—figure supplement 1B*).

## Psychophysiological interaction analysis and correlation with sleep spindles

Psychophysiological interaction (PPI) analysis was computed to test the hypothesis that functional connectivity between the seed region (the right hippocampus seed from the contrast of the test session, *Figure 2C* at (30x, -10y, -23z)) (see Results) and the rest of the brain differed for HR vs. LR hits during the retest session. Therefore, we took as psychological factor the contrast between HR and LR hits, irrespective of trial type (direct, inference 1 and inference 2 trials). A new linear model was prepared for PPI analyses at the individual level, using three regressors. The first regressor represented the psychological factor, composed of HR vs. LR hits. The second regressor was the activity in the right hippocampus. The third regressor represented the interaction of interest between the first (psychological) and the second (physiological) regressor. To build this regressor, the underlying neuronal activity was first estimated by a parametric empirical Bayes formulation, combined with the psychological factor and subsequently convolved with the hemodynamic response function (*Gitelman et al., 2003*). The model also included movement parameters. A significant psychophysiological interaction indicated a change in the regression coefficients between any reported brain area and the reference region, related to the correct retrieval of HR vs. LR trials. Next, individual summary statistic images obtained at the first-level (fixed-effects) analysis were spatially smoothed (6 mm FWHM Gaussian kernel) and entered a second-level (random-effects) analysis using ANOVAs to compare the functional connectivity between groups. Finally, based on existing animal data suggesting a coordinated replay of rewarded associations within striatal and hippocampal regions (*Lansink et al., 2009*), we tested whether reward-related regions showing increased functional connectivity with the hippocampus as a function of reward level (selectively in the Sleep group) correlated with sleep spindles. We thus extracted the beta values around relevant PPI result peaks using a 10 mm diameter sphere and performed a Spearman's rank correlation analysis (appropriate for small samples) with the number of spindles detected during the nap for the participants in the Sleep group.

## Acknowledgements

We are grateful to the Brain and Behaviour Laboratory of the University of Geneva (Geneva, Switzerland) for providing help and scanning facilities. We thank Christoph Hofstetter, Hamdi Eryilmaz and Bruno Bonet for help in operating the fMRI. We thank Neil Burgess and Gethin Hughes for useful comments on the manuscript.

## Additional information

### Funding

| Funder | Grant reference number | Author |
|---|---|---|
| AXA Research Fund | | Kinga Igloi |
| Fondation Fyssen | | Kinga Igloi |
| National Science Foundation | 51NF40-104897 | Sophie Schwartz |
| National Science Foundation | 320030_135653 | Sophie Schwartz |

The funders had no role in study design, data collection and interpretation, or the decision to submit the work for publication.

## Author contributions

KI, SS, Conception and design, Acquisition of data, Analysis and interpretation of data, Drafting or revising the article; GG, VS, Acquisition of data, Analysis and interpretation of data, Drafting or revising the article

## Author ORCIDs

Kinga Igloi, http://orcid.org/0000-0002-3428-8474

## Ethics

Human subjects: All subjects were volunteers, gave written informed consent, consent to publish and received financial compensation for their participation in this study. The study was approved by the Ethics Committee of the Geneva University Hospitals.

# Additional files

## Supplementary files

• Supplementary file 1. Characteristics of the nap period for participants in the Sleep group.

• Supplementary file 2. Whole brain activation table for all reported results.

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
