## [Decision Letter]

Thank you for submitting your work entitled "A nap to recap or how reward regulates hippocampal-prefrontal memory networks during daytime sleep in humans" for peer review at *eLife*. Your submission has been favorably evaluated by Eve Marder (Senior Editor), Heidi Johansen-Berg (Reviewing Editor), and two reviewers.

The reviewers have discussed the reviews with one another and the Reviewing Editor has drafted this decision to help you prepare a revised submission.

This paper investigates how high versus low reward affects sleep-dependent memory consolidation using behavioral methods and fMRI imaging in humans. They show that mainly HR memories benefit from a brief nap after learning. This improvement is correlated with an increase in slow sleep spindles and increased hippocampal activity in fMRI. In a surprise retest after three months, they still find behavioral and fMRI differences between sleep and wake conditions.

Essential revisions:

1) All participants receive the same total payment (first paragraph of the Results section). Thus, the appearance of a reward cue at the onset of learning only provides an indirect predictor of the final outcome, and the final outcome is the same across participants, presumably regardless of performance. Because the total reward amount is presented at the conclusion of the Learning session (Figure 1—figure supplement 1), this post-hoc leveling effect may be apparent to the participants before they go to sleep. This could critically influence the reward effects on subsequent sleep modulation (Figure 1). This needs discussion.

2) In Figure 2 the Group x Reward interaction is not significant, which is a problem for claiming that sleep would selectively enhance high-rewarded memories (see Abstract).

3) Please clarify and justify when correction for multiple comparisons has/has not been applied. For example, in paragraph four of the Results section, the number of spindles correlates with R-hippocampal activation for HR distant associations (P<0.05), but it is unclear whether the authors corrected for multiple comparisons.

4) Claims for specificity need to backed up by direct statistical tests, e.g. in the second and third paragraphs of the Results section, spindles correlate with behavior for HR but not for LR trials. To test whether this relationship is significantly stronger in the HR than the LR trials requires a statistical test such as Fisher’s R to Z.

5) In the Discussion, the conclusion is drawn that slow spindles after learning favor the interplay between HPC and PFC during HR long-term recall, but Figure 3 shows an HPC-lPFC functional connectivity effect without involvement of spindles. Also the PPI analysis (in the subsection “Psychophysiological interaction analysis”) does not include spindles.

6) For interaction effects tested by voxel-wise analyses it is important to clarify what is driving those effects, e.g. Results section, fourth paragraph: the interaction sleep/wake x HR/LR could reflect higher activity in sleep-HR as well as in wake-LR conditions. Please provide bar graphs with actual percent signal changes for all conditions. The same holds for all other interactions (fifth and sixth paragraphs of the Results section).

7) In the Introduction, memory consolidation can also be adaptive if important information is encoded more strongly, and consolidation simply affects everything according to its encoding strength. All results presented in this paper would be similarly predicted if one assumes that strength of memory is determined directly during encoding. The non-significant difference between HR and LR in block 3 seems to come from performance reaching ceiling and does not exclude stronger encoding for HR trials. The Abstract and Discussion should acknowledge that the effects can equally well be explained by stronger encoding of HR items.

8) The paper would benefit from a model or conceptualization in which the functioning of HPC-PFC-VTA networks in the context of the memory task is explained, at least theoretically. An explanatory framework is suggested by the title, but as yet we encounter a number of rather separate findings. As related areas such as N. Accumbens, OFC and Amygdala are part of the same network, how is it explained that no effects were picked up in these areas? Moreover, it remains unclear which mechanisms would account for the fact that sometimes the right hippocampus shows up (e.g. Figure 2), and sometimes the left hippocampus (e.g. 2F – although for Figure 3, the figure legend states "left" hippocampus whereas the text states "right" hippocampus). Likewise, the origin of the lateralization effects is unclear for the PFC.

---

## [Author Response]

1) All participants receive the same total payment (first paragraph of the Results section). Thus, the appearance of a reward cue at the onset of learning only provides an indirect predictor of the final outcome, and the final outcome is the same across participants, presumably regardless of performance. Because the total reward amount is presented at the conclusion of the Learning session (Figure 1), this post-hoc leveling effect may be apparent to the participants before they go to sleep. This could critically influence the reward effects on subsequent sleep modulation (Figure 1). This needs discussion.

We thank you for raising this important point, which indeed requires important clarifications. We told the participants that their performance would directly affect their payment, namely that the dollar sign indicated a sequence that would be more rewarded while the cent indicated a sequence proportionally less rewarded (see Figure 1 and Figure 1—figure supplement 1). Critically, we also explicitly told them that both the dollar and the cent were “symbols” (not actual US dollar values) and that we would convert their performance into Swiss Francs at the end of the learning session, so that if they performed well they would receive the maximal amount (i.e. 120 Swiss Francs). This manipulation was meant to: (i) ensure that the participants would pay attention to the distinct reward levels associated with each sequence, (ii) increase the participants’ motivation to obtain the rewards, and (iii) also ensure that each participant got the same final monetary outcome (i.e. 120 Swiss Francs). Moreover, because the participants could not guess during the learning session how much in Swiss Francs they would receive, we do not think that a post-hoc leveling effect may have occurred. This was confirmed at debriefing, during which participants reported that they believed that they got the maximal amount because they performed well. The revised version of the manuscript now clarifies this important point. More details about this manipulation are provided in the subsection “Behavioural task”.

2) In Figure 2 the Group x Reward interaction is not significant, which is a problem for claiming that sleep would selectively enhance high-rewarded memories (see Abstract).

We realize that the initial formulation in the Abstract may have been misleading. We have adapted the sentence to highlight that the Group x Reward interaction was only significant at retest.

In the new formulation in the Abstract, we show that post-learning sleep favors the selectivity of long-term consolidation: when tested three months after initial encoding, the most important (i.e. rewarded, strongly encoded) memories are better retained, and also remembered with higher subjective confidence.

*3) Please clarify and justify when correction for multiple comparisons has/has not been applied. For example, in paragraph four of the Results section, the number of spindles correlates with R-hippocampal activation for HR distant associations (P<0.05), but it is unclear whether the authors corrected for multiple comparisons.*

We confirm that each reported cluster of activation was corrected for multiple comparisons (familywise error correction) within a priori regions of interest (i.e. small volume correction, SVC). In the revision, we modified the original formulation to clarify this point (third paragraph of the subsection “Functional MRI data acquisition and analysis”).

Concerning the correlation mentioned above by the reviewers, we realize that we did not explicitly justify the use of the contrast “inference 2 > inference 1” for the correlation with sleep spindles. Our aim was to test whether sleep spindles play a critical role in the integration of associative memories. This can be best achieved by contrasting more distant pairs (inference 2 trials) to less distant pairs (inference 1 trials) in inference trials; inference 2 trials involving higher integrative demands than inference 1 trials, while both types of trials have a very similar learning status (unlike direct trials which were first presented (and tested) during the learning session). This contrast should thus reveal brain regions whose activity reflected the strength of the integration between non-consecutive pictures within a series. Therefore, the reported test in the fourth paragraph of the Results section (comparing inference 2 > inference 1) was performed on purpose and does not correspond to one among multiple possible options. We have now clarified this point in the revised version of the manuscript.

*4) Claims for specificity need to backed up by direct statistical tests, e.g. in the second and third paragraphs of the Results section, spindles correlate with behavior for HR but not for LR trials. To test whether this relationship is significantly stronger in the HR than the LR trials requires a statistical test such as Fisher’s R to Z.*

We thank you for the suggestion and have performed a Fisher’s R to Z test adapted for two dependent correlations with one variable in common (here slow spindles) (Lee and Preacher, 2013) and report a significant difference between the correlations for HR and LR trials (z-score=2.15, two-tailed p=0.01). This new result is now included in the revised version of the manuscript, in the third paragraph of the Results section.

*5) In the Discussion, the conclusion is drawn that slow spindles after learning favor the interplay between HPC and PFC during HR long-term recall, but Figure 3 shows an HPC-lPFC functional connectivity effect without involvement of spindles. Also the PPI analysis (in the subsection “Psychophysiological interaction analysis”) does not include spindles.*

We agree that this claim was not directly backed up by our analyses. We have now clarified the way the PPI was done and conducted the PPI analysis without masking as asked by the reviewer(s) for the other contrasts. This new analysis yielded two significant activation clusters, one in the medial prefrontal cortex [z-score=2.99 (15x, 59y, -5z); <0.05 SVC] and one in the caudate nucleus [z-score=3.41 (15x, 20y, 7z); p<0.05 SVC). To directly address the reviewers’ comment, and because the striatal-hippocampal coupling may be relevant for the selective replay of rewarded associations during sleep (Lansink et al., 2009), we extracted the beta values of the PPI around the caudate nucleus peak using a 10 mm diameter sphere and correlated these values to the number of slow sleep spindles during the nap in the sleep group. We performed a Spearman’s rank correlation (appropriate for small samples, here n=12) and obtained a significant correlation (R=0.591,p=0.028). Note that no such correlation was observed for the connectivity between the hippocampus and mPFC. From these connectivity findings, we conclude that slow spindles may primarily favor the long-term consolidation of rewarded stimuli across striatal and hippocampal networks. We may speculate that enhanced hippocampal recruitment for HR trials at retest could possibly foster recall processes subserved by the mPFC, a brain region reportedly involved in the retrieval of episodic memories (Duzel et al., 1999).The new correlation has been added to Figure 3 and to the manuscript in paragraph five Results section. The interpretation of the results has been adapted to account for these new data (second paragraph of the Discussion).

*6) For interaction effects tested by voxel-wise analyses it is important to clarify what is driving those effects, e.g. Results section, fourth paragraph: the interaction sleep/wake x HR/LR could reflect higher activity in sleep-HR as well as in wake-LR conditions. Please provide bar graphs with actual percent signal changes for all conditions. The same holds for all other interactions (fifth and sixth paragraphs of the Results section).*

We have added the requested bar graphs to the figures. For all four interactions, post-hoc analyses on the parameter estimates of signal change across conditions revealed that activity was systematically higher for the HR vs. LR trials for the Sleep group, while the opposite difference (LR vs. HR) in the Wake group was only significant for the main effect of sleep on reward at test (Figure 2). This general pattern of results suggests that it is the differential impact of sleep on memory and reward circuits as a function of reward values that drives the interaction effects. These new results are added to the revised manuscript (Results and Materials and methods sections).

*7) In the Introduction, memory consolidation can also be adaptive if important information is encoded more strongly, and consolidation simply affects everything according to its encoding strength. All results presented in this paper would be similarly predicted if one assumes that strength of memory is determined directly during encoding. The non-significant difference between HR and LR in block 3 seems to come from performance reaching ceiling and does not exclude stronger encoding for HR trials. The Abstract and Discussion should acknowledge that the effects can equally well be explained by stronger encoding of HR items.*

We agree with the reviewers that, beyond ‘pure’ reward effects, encoding strength might also influence consolidation processes. To minimize possible effects of encoding disparities due to reward levels, in all the analyses and for each participant, we only included well encoded items, namely associations for which the participant was correct on at least 2 over 3 repetitions during the learning phase. To directly address the reviewers’ comment, we have now performed a new analysis for the test session, further separating items according to their encoding strength (i.e. separating items for which a participant gave 2 or 3 correct responses during learning) with reward, inference and group as other factors. This analysis revealed no effect of encoding strength at test (F(1,74)=0.0002, p>0.05) neither any interaction effect with encoding strength (all ps>0.05), thus suggesting that associated reward value rather than stronger encoding promotes sleep-related memory consolidation in our study. Consistent with our results, one previous study which specifically tested for the effects of encoding strength on sleep-related memory consolidation found that sleep favors the consolidation of less well encoded material (more than thoroughly encoded material) (Schmidt et al., 2006). Concerning the specific effect of reward, one behavioral study in which instruction about the future relevance was provided after an encoding session, also demonstrated that sleep selectively promotes the consolidation of information associated with a positive future outcome (for information with a similar level of initial encoding) (Fischer and Born, 2009). Moreover, animal data showing that neural activity recorded during a place-reward associative learning task may be reactivated (and possibly consolidated) during sleep across both reward and memory systems (Lansink et al., 2009) may also point to a specific role of associated reward value for the reprocessing of hippocampal-dependent memories during sleep. Although these considerations and the new analysis reported above support our initial interpretation that highly (compared to lowly) rewarded items benefit more from consolidation processes occurring during sleep, we now acknowledge in the Abstract and Discussion that encoding strength may also have an influence on these offline consolidation processes, despite our efforts to minimize disparities in encoding strength.

*8) The paper would benefit from a model or conceptualization in which the functioning of HPC-PFC-VTA networks in the context of the memory task is explained, at least theoretically. An explanatory framework is suggested by the title, but as yet we encounter a number of rather separate findings. As related areas such as N. Accumbens, OFC and Amygdala are part of the same network, how is it explained that no effects were picked up in these areas? Moreover, it remains unclear which mechanisms would account for the fact that sometimes the right hippocampus shows up (e.g. Figure 2), and sometimes the left hippocampus (e.g. 2F – although for Figure 3, the figure legend states "left" hippocampus whereas the text states "right" hippocampus). Likewise, the origin of the lateralization effects is unclear for the PFC.*

We thank the reviewers for this suggestion, which prompted us to propose a descriptive model that would incorporate the main findings of our study to offer an integrative account of how rewards may act as a “relevance tag” that would prioritize the neural reprocessing of associative memories during sleep. Specifically, our reward-based learning paradigm implicates distinct mechanisms that can best be described as a 2-step process. (1) At encoding, (high) potential rewards activate dopaminergic midbrain regions, which would seal a relevance tag to recently learned and rewarded associations. This initial step is substantiated by our data showing that VTA activity at encoding mediates the acquisition of highly rewarded material. (2) During sleep, the privileged reprocessing of associations with a high relevance would plausibly implicate the activation of a VTA–striatum-hippocampal network (Lansink et al., 2009; Perogamvros and Schwartz, 2012; Valdes et al., 2015), and be coordinated by oscillatory activity during sleep, in particular sleep spindles (Born et al., 2006; Molle et al., 2011; Schmidt et al., 2006; Wilhelm et al., 2011), during which the human VTA has been found to be activated (Schabus et al., 2007). Here, we report enhanced performance and higher hippocampus activity for the recall of highly rewarded associations. Critically, although we did not acquire fMRI data during sleep in our study, we also show that the number of slow spindles correlated with the selective consolidation of highly rewarded associations at test and, at long-term, with increased hippocampal-striatal connectivity, thus consistent with a determinant role of sleep spindles in the consolidation of rewarded associations across memory and reward brain regions.

Together with increased connectivity between the hippocampus and the medial prefrontal activity at long-term retest, our findings also fit the general theoretical framework according to which memories are stored in the medial prefrontal cortex (Takashima et al., 2009), but may recruit the hippocampus for recall (Harand et al., 2012). In the case of rewarded memories the striatum may have an important contribution (Wittmann et al., 2005), in line with recent observations of joint caudate and prefrontal involvement during the recall of positive relative to neutral autobiographical memories in humans (Speer et al., 2014).

We have now added part of this answer to the Discussion section.

Concerning hippocampal activations, we only report activations surviving a threshold of p<0.001. However, subthreshold peaks (p<0.05) are detected in the contralateral hippocampus for all reported contrasts (please see [Supplementary-material SD2-data]). In the present study, we did not have any a priori hypothesis concerning the lateralization of hippocampal activity because our task involves associative memory while the nature of our stimuli involves the processing of spatial contexts, which may respectively recruit the left and the right hippocampus (Igloi et al., 2010).